# Epidemiology and molecular characterisation of multidrug-resistant *Escherichia coli* isolated from chicken meat

Hamida Khanom[1,2], Chandan Nath[1], Philip P. Mshelbwala[3], Md Ridoan Pasha[1], Ricardo Soares Magalhaes[4,5], John I. Alawneh[6], Mohammad Mahmudul Hassan[1,4]*

**1** Faculty of Veterinary Medicine, Chattogram Veterinary and Animal Sciences University, Chattogram, Bangladesh, **2** Department of Biosciences and Biotechnology, Fukui Prefectural University, Yoshida-gun 910, Fukui, Japan, **3** Department of Primary Industries, Orange, New South Wales, Australia, **4** Queensland Alliance for One Health Sciences, School of Veterinary Science, The University of Queensland, Gatton, Queensland, Australia, **5** Children's Health and Environment Program, Children Health Research Centre, The University of Queens-land, Brisbane, Queensland, Australia, **6** Plant Biosecurity and Product Integrity, Biosecurity Queensland, Department of Primary Industries, Brisbane, QLD 4000, Australia.

* miladhasan@cvasu.ac.bd

## Abstract

Ensuring the safety of poultry products is critical for public health, particularly due to the rising concern of antimicrobial resistance (AMR) in foodborne pathogens. This study aimed to investigate the prevalence and antimicrobial resistance (AMR) patterns of *Escherichia coli (E. coli)* isolated from broiler chicken meat samples collected from live bird markets (LBMs) and supermarkets (SMs) in the Chattogram Metropolitan Area (CMA), Bangladesh. A total of 430 samples, comprising 215 liver and 215 muscle samples, were collected between October 2020 and February 2021 from nine LBMs and five SMs. Samples were processed and cultured, and *E. coli* was isolated and identified through phenotypic and molecular techniques, including PCR targeting the *uid*A and *usp*A genes. Antimicrobial susceptibility testing (AST) was conducted using the Kirby-Bauer disk diffusion techniques with seven antibiotics from six distinct antimicrobial classes. The study found an overall prevalence of 56.28% (95% CI: 51.56–60.89) for *E. coli*. The prevalence in LBMs (58.33%) was higher than in SMs (54.80%), with liver samples showing a slightly higher rate of contamination (63.33% in LBMs, 55.20% in SMs) compared to muscle samples. AMR profiling revealed high resistance rates to sulfamethoxazole-trimethoprim (88.84%), tetracycline (86.78%), and ampicillin (82.23%). Conversely, cephalexin (63.64%) and gentamicin (57.02%) had the highest susceptibility rates. A significant proportion (84.71%) of isolates were multidrug-resistant (MDR), with some isolates resistant to up to six classes of antimicrobial. The multiple antibiotic resistance (MAR) index ranged from 0.14 to 1.00, indicating substantial antimicrobial exposure. PCR analysis confirmed the presence of the *bla*TEM gene in all ampicillin-resistant isolates, while 75.35%

**Data availability statement:** All relevant data are within the article and its Supporting Information files.

**Funding:** Bangladesh Bureau of Education Information and Statistics (BANBEIS), Ministry of Education, People's Republic of Bangladesh, funded this research with project number #SD-2019967.

**Competing interests:** The authors have declared that no competing interests exist.

of sulfamethoxazole-resistant isolates carried the *sul*2 gene. Correlation analysis revealed a strong association between phenotypic resistance to ampicillin and the presence of the *bla*TEM gene (r = 1), along with a moderate correlation between *sul2* and resistance to sulfamethoxazole (r = 0.5). These findings highlight the widespread presence of multidrug-resistant (*MDR*) *E. coli* in broiler meat, posing a significant public health concern.

## Introduction

*Escherichia coli (E. coli)* is a bacterium essential for supporting intestinal health in both humans and animals [1]. Although the majority of *E. coli* strains are non-pathogenic, approximately 10–15% of intestinal coliforms comprise opportunistic and pathogenic serotypes capable of causing infections in immunocompromised hosts, including poultry [2]. Contamination of meat with *E. coli* is frequently associated with poor slaughter hygiene [3], and the strains isolated from such contaminated meat have shown resistance to commonly used antibiotics [4]. This antimicrobial resistance presents serious threats to the health of both humans and animals [5]. While the majority of *E. coli* strains coexist harmlessly in the large intestine, they can become pathogenic under specific conditions, causing both intestinal and systemic infections [6]. Antimicrobials are extensively used in both human and veterinary medicine to manage infections and health risks, but their overuse contributes to the rise of AMR, posing a serious challenge to effective treatment and public health [7]. Antimicrobials are often administered as growth promoters (AMGP) in poultry farming, especially for broilers. However, to minimise the risk of resistance development, they should be used cautiously and primarily for therapeutic and preventive purposes [8]. Overuse of antibiotics accelerates the development of antibiotic-resistant bacteria, a complex process driven by bacterial genetic and metabolic mechanisms [9,10].

In intensive broiler chicken production, the strong antibiotic selection pressure has led to a substantial presence of resistant bacteria in poultry fecal flora, which can then spread to humans, pets, and the environment [11–13]. The misuse of antimicrobials in livestock production contributes to developing multidrug resistant *E. coli*, which can be spread to humans via the food chain, particularly through broiler chicken meat, posing serious health risks as well as making treatment option more challenging [14,15].

LBMs and SMs play an important role for influencing microbial contamination and AMR patterns due to their diverse poultry supply chain, meat processing methods and consumer exposure risks [15]. Poor handling, inadequate cleaning, and improper meat-selling practices are known to contribute to poultry meat contamination with *E. coli* [15]. Poultry meat, particularly breast muscle and liver of broiler chicken, has been recognised as a likely source of infection due to the high risk of contamination during meat processing [16]. While the global prevalence of avian *E. coli* in broiler meat has been well documented, there is still limited information on the situation in Bangladesh. Despite the presence of veterinary and research laboratories across

Bangladesh, systematic surveillance of AMR remains inadequate. Over 60% of farmers in Bangladesh reportedly use anti-biotics without prescriptions [17]. Chicken meat from LBMs has AMR, as vendors often use antibiotics to reduce mortality rates among their stock [18]. Moreover, a range of antimicrobials, including ciprofloxacin, amoxicillin, sulfamethoxazole-trimethoprim, and gentamicin, are frequently used at various stages of poultry production, contributing to the development of antibiotic-resistant *E. coli* strains in Bangladesh. The rate of antimicrobial resistance have increased at a faster rate in *E. coli* as well as commensal *E. coli* which is an important reservoir of antimicrobial resistance genes [19].These antimicrobial resistance genes may disseminate to pathogenic strain through horizontal gene transfer mechanism because *E. coli* has ability to get mobile genetic elements including plasmids, transposons, and integrons, which often carry AMR genes [20]. Polymerase chain reaction (PCR), DNA microarrays, and whole-genome sequencing (WGS) are all performed for molecular characterization, which is crucial for precisely identifying the genetic determinants of drug resistance bacteria. These technologies improve our understanding of the genetic basis of resistance and pathogenicity, allowing for more efficient surveillance and control efforts. Several studies on AMR *E. coli* on chicken meat in Bangladesh have been published previously [15,18,21], but this study focuses on new insights such as comparative analysis Between LBMs and SMs, correlation of resistance genes with phenotypic resistance, and molecular characterization. This study examines the current levels of *E. coli* contamination and AMR profiles, including *bla*TEM, *sul*1, and *sul*2 genes, in broiler chicken meat from SMs and LBMs in the Chattogram Metropolitan Area (CMA) of Bangladesh.

## Methodology

### Ethical approval

This study was conducted by following the Declaration of Helsinki, and the protocol was approved by the Ethics Committee of the Chattogram Veterinary and Animal Sciences University, Bangladesh (permit reference number: CVASU/Dir (R and E) EC/2019/126 (02), Date: 29 December 2019).

### Study location and design

This cross-sectional study was carried out between October 2020 and February 2021 in the Chattogram Metropolitan Area (CMA) of Bangladesh's second-largest city. There are numerous LBMs and SMs in the study region. The LBM stalls are within proximity, which increases the risk of cross contamination [15]. This research is a part of a larger project, with some components already published [18,21]. A total of 40 LBMs and eight SMs, with nine LBMs and five SMs conveniently chosen for sample collection within the CMA. Ten stalls were randomly selected from each LBM. We collected breast muscle and liver tissues, as these are primary sites for bacterial contamination during slaughter and are commonly consumed, directly impacting public health. The geographical locations of the sampling areas are shown in Fig 1.

### Sample collection, transportation, and processing

We determined the appropriate sample size using the Open Epi version 3.1 online tool, and based on the following equation

$$\text{Sample size } (n) = [DEFF \times Np(1-p)]/[(d2/Z21 - \alpha/2 \times (N-1) + p \times (1-p)]$$

Based on a prior study [18], in which the design effect was fixed at 1, we estimated the expected frequency of the outcome factor in the population (p) to be 76.1% +/- 5% error [18].

From each LBM, 10 liver samples and 10 breast muscle samples were collected; from each SM, 25 liver samples and 25 muscle samples were collected. In total, 430 samples were obtained, consisting of 215 liver and 215 muscle samples. In the SMs, the sources of chickens were from different farms and LBMs. Usually, SMs collected poultry from different suppliers throughout the years. The broiler carcasses were kept at $4^{0}$C overnight and sometimes until sold out.

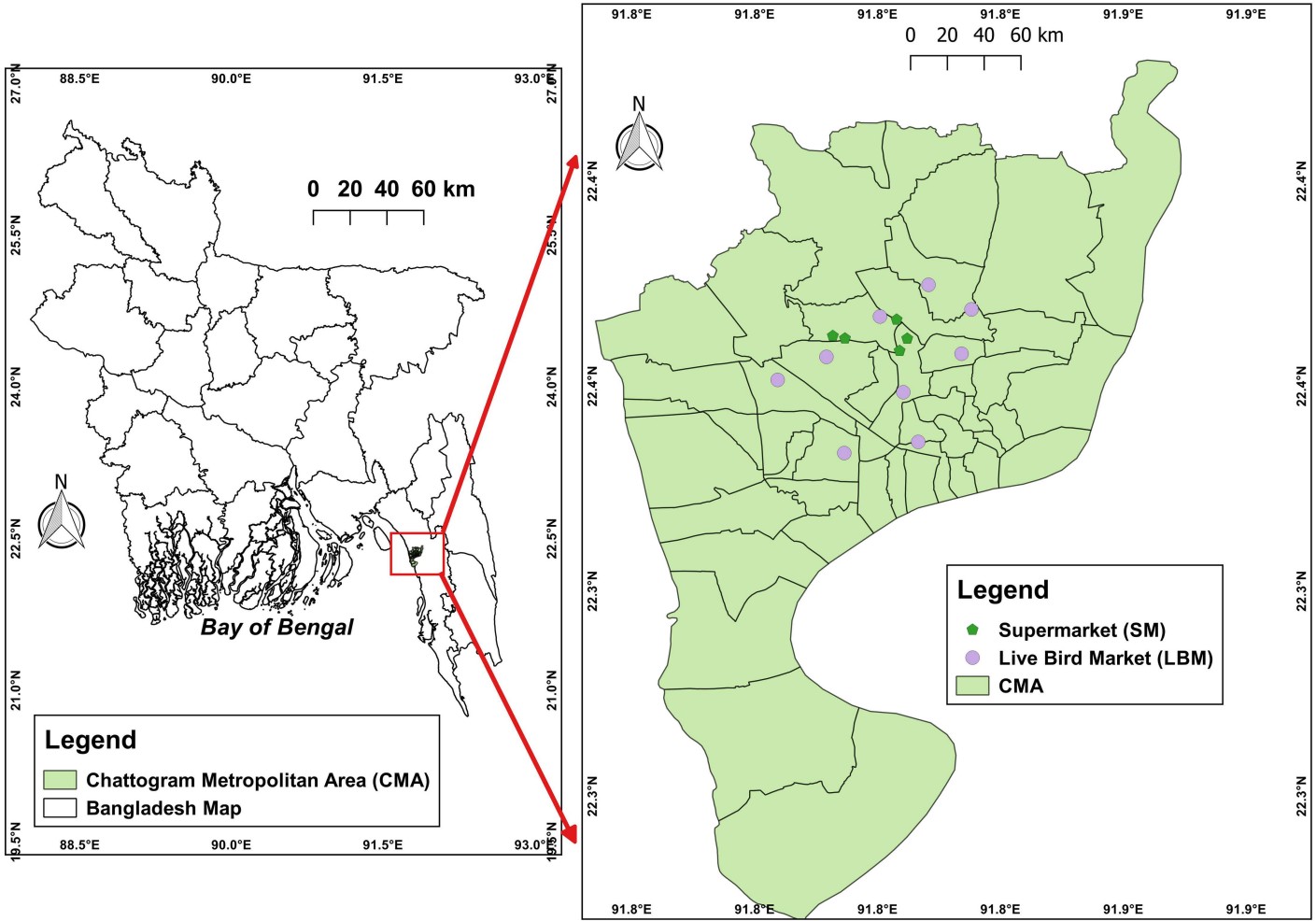

**Fig 1. Map depicts locations of selected LBMs and SMs of Chattogram Metropolitan Area.**

An overview of the sample collection is presented in S1 Table. The samples were collected in separate zipper bags while maintaining proper hygiene procedures. In brief, one dressed broiler carcass was collected from each stall. After collection, they were transported to the Department of Microbiology and Veterinary Public Health (DMVPH) at Chattogram Veterinary and Animal Sciences University (CVASU) for further investigation, ensuring the cold chain was maintained throughout transport. The samples were then processed into small pieces using sterile scissors, and 1 g of each sample was transferred into a separate sterile Falcon tube containing 9 mL of buffered peptone water (BPW) (HIMEDIA, pH: 7.0 ± 0.2, Mumbai, India). The samples were incubated at 37°C overnight for primary enrichment.

## Isolation and presumptive detection of *E. coli*

To isolate *E. coli*, the enriched culture was streaked onto MacConkey agar medium (HIMEDIA, pH: 7.1 ± 0.2, Mumbai, India) and incubated at 37°C for 24 hours. Bright pink-coloured, large colonies on the MacConkey agar plate were suspected of *E. coli*. These colonies were then streaked onto Eosin Methylene Blue (EMB) agar plates (HIMEDIA, pH: 7.0 ± 0.2, Mumbai, India) and incubated at 37°C for 24 hours. A "green metallic sheen" on the EMB agar confirmed the growth of *E. coli*. Following confirmation, the isolates were inoculated onto blood agar (HIMEDIA, pH: 7.0 ± 0.2, Mumbai,

India) and incubated at 37°C for 24 hours. All confirmed *E. coli* isolates were cultured in brain heart infusion (BHI) broth (HIMEDIA, pH: 7.0±0.2, Mumbai, India) and incubated overnight at 37°C. For each isolate, 700 μL of the BHI broth culture was mixed with 300 μL of 15% glycerol in an Eppendorf tube. The tubes were properly labelled and stored at -80°C for further investigation.

## Molecular detection of *E. coli*

Phenotypic *E. coli* isolates were subjected to molecular identification by PCR. Genomic DNA was extracted using the crude boiling method described by Malorny et al. [22]. Molecular identification of *E. coli* was carried out through species-specific multiplex PCR in a thermal cycler (DLAB, USA) using primers targeting the *uid*A gene and the flanking region of the *usp*A gene [23]. The oligonucleotide primer sequences are listed in Table 1. The PCR reaction mixture consisted of 12.5 μL of OneTaq Quick-load 2X Master Mix with Standard Buffer (New England Biolabs Inc.), 0.5 μL each of the forward and reverse primers (10 pmol), 1 μL of template DNA, and the required volume of nuclease-free water. The thermal cycling conditions were as follows: initial denaturation at 94°C for 5 minutes, followed by 35 cycles of denaturation at 94°C for 10 seconds, annealing at 55.2°C for 10 seconds, extension at 72°C for 1 minute, and a final extension at 72°C for 10 minutes. After amplification, 5 μL of the PCR products were loaded onto a 1.5% (w/v) agarose gel, prepared using agarose powder (MP Biomedicals, USA) and 1X TAE buffer (Thermo Fisher Scientific, USA). The gel was visualised in a gel documentation system (UVP UVsolo touch, Analytik Jena AG, Thermo Fisher Scientific, USA) after staining with ethidium bromide (Sigma-Aldrich, USA).

## AST of *E. coli*

The *E. coli* positive isolates in PCR were screened for antimicrobial susceptibility against a panel of antimicrobials using the Kirby-Bauer disc diffusion method [24]. Seven antimicrobials of six different groups such as penicillins: ampicillin, tetracyclines: tetracycline and doxycycline, aminoglycosides: gentamicin, fluoroquinolones: ciprofloxacin, sulfonamides: sulfamethoxazole-trimethoprim, and cephalosporins: cephalexin) of drugs that had public health significance were selected for antimicrobial susceptibility testing. All the types, i.e., Access, Watch or reserve groups of antibiotics, were available in the study area, but the farmers did not follow the policy for antibiotic use among poultry farms. The following antimicrobials with respective disc potencies were used: TE: tetracycline (30μg), CN: gentamicin (10μg), DO: doxycycline (30μg), AMP: ampicillin (10μg), CL: cephalexin (30μg), SXT: sulfamethoxazole-trimethoprim (23.75μg + 1.25 μg), CIP: ciprofloxacin (5μg). After dispensing all the discs, the Mueller Hinton agar (HIMEDIA, pH: 7.0±0.2, Mumbai, India) plates were incubated at 37°C for 18 hours. After incubation, the size of the zone of inhibition (in mm) around a disc, including

**Table 1. The oligonucleotide primer sequences used in this study.**

| Target gene | Primer Name | Primer sequence (5′-3′) | Annealing temperature | Amplicon size (bp) | References |
|---|---|---|---|---|---|
| *usp*A | *usp*A -F | CCGATACGCTGCCAATCAGT | 55.2 °C | 884 | [23] |
| | *usp*A -R | ACGCAGACCGTAGGCCAGAT | | | |
| *uid*A | *uid*A-F | TATGGAATTTCGCCGATTTT | | 164 | |
| | *uid*A -R | TGTTTGCCTCCCTGCTGCGG | | | |
| *bla*TEM | *bla*TEM - F | GCGGAACCCCTATTTG | 50 °C | 964 | [29] |
| | *bla*TEM - R | TCTAAAGTATATATGAGTAAACTTGGTCTGAC | | | |
| *sul* 1 | *sul* 1- F | GTGACGGTGTTCGGCATTCT | 68 °C | 779 | [28] |
| | *sul* 1- R | TCCGAGAAGGTGATTGCGCT | | | |
| *sul* 2 | *sul* 2- F | CGGCATCGTCAACATAACCT | 66 °C | 721 | |
| | *sul* 2- R | TGTGCGGATGAAGTCAGCTC | | | |

the diameter of the disc, was measured using slide calipers, and the result was interpreted according to Clinical Laboratory Standards Institute guidelines [25]. The pan-susceptible *E. coli* ATCC 25922 was used as a quality control strain during AST. The *E. coli* isolates resistant to at least three antimicrobial classes were described as multidrug-resistant (MDR) [26]. The multiple antibiotic resistance (MAR) index was estimated by following the formula a/b, a = the number of antibiotics that showed resistance by isolates, b = the number of antibiotics that were exposed by isolates, as described earlier by Krumperman et al., [27].

### Detection of antimicrobial resistance genes

Using PCR, the phenotypic sulphonamide- and ampicillin-resistant isolates were tested for the presence of resistance genes. The *sul* 1 and *sul* 2 genes were screened for sulphonamide resistance, and the *bla*TEM gene for ampicillin resistance, as previously described by Lanz et al. [28] and Hasman et al. [29]. The oligonucleotide primer sequences, amplicon sizes, and annealing temperatures are provided in Table 1.

### Statistical analysis

All sampling and laboratory test data were meticulously entered, organised, and cleaned using Microsoft Excel 2019. Descriptive statistics, including percentages and 95% confidence intervals, were performed using the modified Wald method available in the GraphPad QuickCalcs online tool (https://www.graphpad.com/quickcalcs/). Pearson's Chi-squared test was performed on LBMs and SMs to determine the level of significance of the source. The *p* value ≤0.05 was considered as statistically significant. Data visualisations, such as heat maps and bar charts, were generated using GraphPad Prism 7.0 (GraphPad Software, La Jolla, CA, USA) to illustrate trends and distributions effectively. Correlation analyses between antimicrobials and corresponding resistance gene abundances were conducted using R software (version 4.4.1; https://www.r-project.org/). These correlations were visualised with high-quality graphical outputs produced in R, ensuring precise and informative data presentation. The map of the locations of the LBMs and SMs was created using QGIS software (version 3.12.0).

## Results

### Prevalence of *E. coli* in LBMs and SMs

Among the 450 samples tested in this study, 242 (56.28%, 95% CI: 51.56–60.89) were confirmed as *E. coli*. In LBMs, the prevalence of *E. coli* was 58.33%, with 63.33% in liver and 52.22% in muscle samples (S2 Table). Similarly, the prevalence in SMs was 54.80%, with 55.20% in liver samples and 54.40% in muscle samples. The prevalence of *E. coli* across different sources is presented in Table 2.

### AMR patterns of *E. coli* isolates of LBMs and SMs

The AST showed that the highest proportion of isolates were resistant to sulfamethoxazole-trimethoprim (88.84%), followed by tetracycline (86.78%) and ampicillin (82.23%). In contrast, the highest susceptibility was confirmed for cephalexin (63.64%), followed by gentamicin (57.02%) and ciprofloxacin (23.14%). Additionally, 28.51% of the isolates were intermediate to doxycycline. The AMR profiles are illustrated in Fig 2/a.

### MDR profiles of *E. coli*

This study revealed that 205 isolates (84.71%) exhibited multidrug resistance (MDR), while 37 isolates (15.29%) were not MDR. Among the non-MDR isolates, 3 (1.24%) showed no resistance to any antimicrobials tested. Of the MDR isolates, 78 (32.23%) were resistant to four antimicrobial classes (Fig 2/b). The phenotypic antimicrobial resistance patterns revealed that 32 isolates (13.22%) were resistant to four antimicrobial classes, displaying the CIP + SXT + AMP + DO + TE resistance patterns. Additionally, 25 isolates (10.33%) exhibited resistance to five antimicrobial classes, displaying the

**Table 2. The overview of the sample collected and frequency of *E. coli* from different sources.**

| Sources | Samples | N, *E. coli* | Percentage (%), (95% CI) | Chi-Square, *p* value |
|---|---|---|---|---|
| LBMs | Liver (n = 90) | 57 | 63.33 (53.01-72.57) | 0.53098 |
|  | Muscle (n = 90) | 48 | 52.22 (42.02-62.24) |  |
| subtotal (n = 180) |  | 105 | 58.33 (51.03-65.29) |  |
| SMs | Liver (n = 125) | 69 | 55.20 (46.46-63.63) | 0.76[ns*] |
|  | Muscle (n = 125) | 68 | 54.40 (45.67- 62.87) |  |
| subtotal (n = 250) |  | 137 | 54.80 (48.60-60.85) |  |
| Total (N = 430) |  | 242 | 56.28 (51.56-60.89) |  |

[*]Not significant

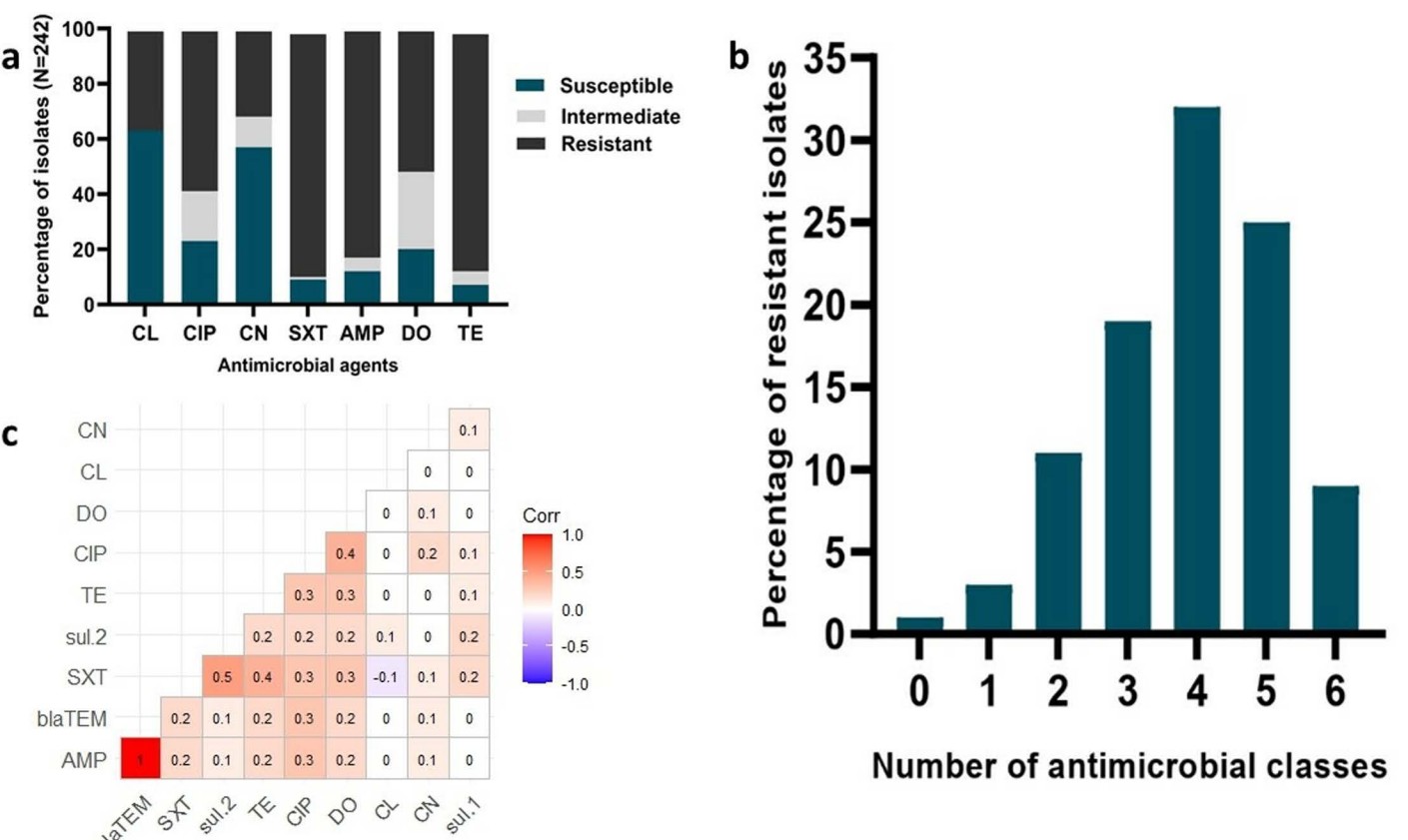

**Fig 2. AMR patterns of E. coli isolated from broiler chicken muscle and liver samples. a) AMR patterns, b) MDR patterns and c) correlation coefficient of specific antimicrobials and resistance genes.** Here, TE = tetracycline, DO = doxycycline CN = gentamicin, CIP = ciprofloxacin, SXT = sulfamethoxazole-trimethoprim, CL = cephalexin and AMP = ampicillin.

CIP + CN + SXT + AMP + DO + TE resistance pattern. Furthermore, 15 isolates (6.22%) demonstrated resistance to six anti-microbial classes following the CL + CIP + CN + SXT + AMP + DO + TE resistance pattern (Table 3). The multiple antibiotic resistance (MAR) index for all isolates ranged from 0.14 to 1 (Table 3).

**Table 3. The phenotypic AMR patterns of *E. coli* isolates, along with the MAR index.**

| No of isolates | % | Resistance type | Phenotypic resistance patterns | MAR Index |
|---|---|---|---|---|
| 32 | 13.22 | MDR | *Four classes*<br>CIP, SXT, AMP, DO, TE | 0.57 |
| 25 | 10.33 | MDR | *Five classes*<br>CIP, CN, SXT, AMP, DO, TE | 0.86 |
| 18 | 7.44 | MDR | *Five classes*<br>CL, CIP, SXT, AMP, DO, TE | 0.86 |
| 18 | 7.44 | MDR | *Three classes*<br>SXT, AMP, TE | 0.43 |
| 15 | 6.20 | MDR | *Six classes*<br>CL, CIP, CN, SXT, AMP, DO, TE | 1 |
| 12 | 4.96 | MDR | *Four classes*<br>CIP, SXT, AMP, TE | 0.57 |
| 8 | 3.31 | MDR | *Three classes*<br>SXT, AMP, DO, TE | 0.57 |
| 7 | 2.89 | MDR | *Four classes*<br>CL, SXT, AMP, TE | 0.57 |
| 7 | 2.89 | MDR | *Five classes*<br>CIP, CN, SXT, AMP, TE | 0.71 |
| 6 | 2.48 | MDR | *Five classes*<br>CL, CIP, SXT, AMP, TE | 0.71 |
| 6 | 2.48 | MDR | *Six classes*<br>CL, CIP, CN, SXT, AMP, DO | 0.86 |
| 5 | 2.07 | MDR | *Four classes*<br>CL, SXT, AMP, DO, TE | 0.57 |
| 4 | 1.65 | MDR | *Four classes*<br>CN, SXT, AMP, TE | 0.57 |
| 3 | 1.24 | MDR | *Four classes*<br>SXT, AMP, CIP, TE | 0.57 |
| 3 | 1.24 | MDR | *Three classes*<br>CL, AMP, TE | 0.43 |
| 3 | 1.24 | MDR | *Three classes*<br>CL, SXT, TE | 0.43 |
| 2 | 0.83 | MDR | *Five classes*<br>CL, CN, SXT, AMP, DO, TE | 0.86 |
| 2 | 0.83 | MDR | *Five classes*<br>CL, CN, SXT, AMP, TE | 0.71 |
| 2 | 0.83 | MDR | *Four classes*<br>CL, CIP, AMP, TE | 0.57 |
| 2 | 0.83 | MDR | *Four classes*<br>CL, CN, SXT, TE | 0.57 |
| 2 | 0.83 | MDR | *Four classes*<br>CIP, CN, SXT, AMP | 0.57 |
| 2 | 0.83 | MDR | *Three classes*<br>CIP, SXT, TE | 0.43 |
| 2 | 0.83 | MDR | *Three classes*<br>CIP, SXT, DO, TE | 0.57 |
| 1 | 0.41 | MDR | *Three classes*<br>CL, AMP, TE | 0.43 |
| 1 | 0.41 | MDR | *Three classes*<br>CL, SXT, AMP | 0.43 |

*(Continued)*

**Table 3.** (Continued)

| No of isolates | % | Resistance type | Phenotypic resistance patterns | MAR Index |
|---|---|---|---|---|
| 1 | 0.41 | MDR | *Three classes*<br>CL, CIP, AMP | 0.43 |
| 1 | 0.41 | MDR | *Three classes*<br>CL, SXT, DO, TE | 0.57 |
| 1 | 0.41 | MDR | *Three classes*<br>CIP, CN, SXT | 0.43 |
| 1 | 0.41 | MDR | *Three classes*<br>SXT, AMP, DO | 0.43 |
| 1 | 0.41 | MDR | *Three classes*<br>CL, SXT, DO | 0.43 |
| 1 | 0.41 | MDR | *Three classes*<br>CL, CIP, TE | 0.43 |
| 1 | 0.41 | MDR | *Three classes*<br>CN, SXT, AMP | 0.43 |
| 1 | 0.41 | MDR | *Three classes*<br>CIP, CN, SXT, TE | 0.57 |
| 1 | 0.41 | MDR | *Four classes*<br>CL, CN, SXT, AMP | 0.57 |
| 1 | 0.41 | MDR | *Four classes*<br>CIP, SXT, AMP, DO | 0.57 |
| 1 | 0.41 | MDR | *Four classes*<br>CL, CIP, SXT, TE | 0.57 |
| 1 | 0.41 | MDR | *Four classes*<br>CL, CN, SXT, DO | 0.57 |
| 1 | 0.41 | MDR | *Four classes*<br>CL, CIP, SXT, DO, TE | 0.71 |
| 1 | 0.41 | MDR | *Four classes*<br>CN, SXT, AMP, DO, TE | 0.71 |
| 1 | 0.41 | MDR | *Four classes*<br>CL, CN, SXT, DO, TE | 0.71 |

### Correlation of antimicrobials, resistance genes with *E. coli* isolates from LBMs and SMs

The Pearson's Correlation coefficient analysis revealed a strong positive correlation between AMP resistance and the *bla*TEM gene ($r = 1$), a moderate correlation between SXT resistance and the *sul*2 gene ($r = 0.5$), and a weak correlation between SXT resistance and the *sul*1 gene ($r = 0.2$). The weak correlation ($r = 0.2$) between *bla*TEM and SXT, TE, or *sul*2 indicates an indirect or minimal relationship (Fig 2/c). The heat map compares the AMR profiles of liver and meat samples collected from LBMs and SMs alongside the *sul*1, *sul*2, and *bla*TEM genes associated with sulfonamide and ampicillin resistance (Fig 3).

## Discussion

AMR is an escalating global issue as the effectiveness of antimicrobials continues to decline. The results of this study demonstrated a high prevalence of *E. coli* in broiler breast muscle and liver samples from both SMs and LBMs, with these isolates showing resistance to multiple antimicrobials.

In this study, the prevalence of *E. coli* in broilers from LBMs is consistent with the findings of Hossain et al. [30], who reported a prevalence of 63.6%; Jakaria et al. [31], who found 82%, and Bashar et al. [32], who reported 100% in poultry. The ampicillin resistance observed in this study is consistent with the findings of Islam et al. [33], who reported a 100%

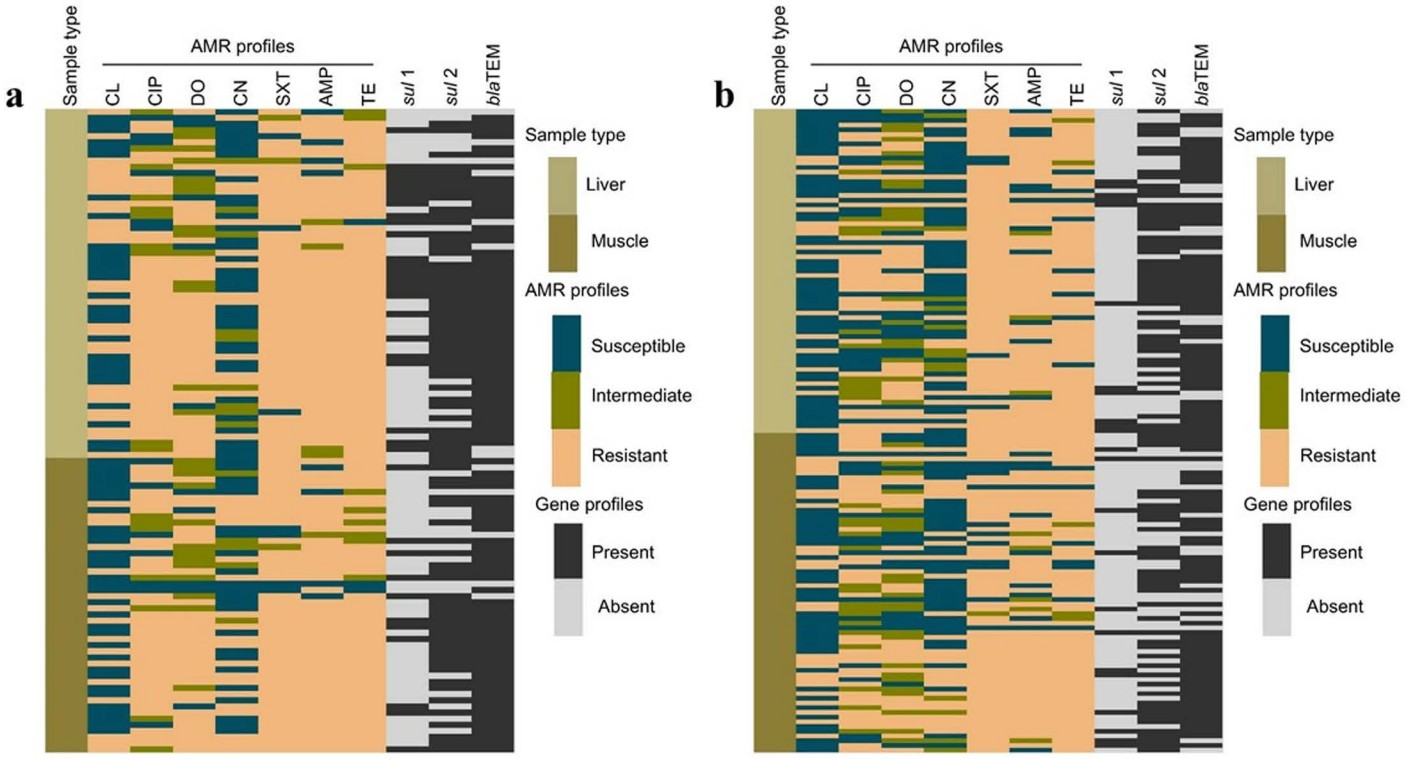

**Fig 3. The heatmap illustrates the AMR profiles of liver and muscle samples, along with resistance genes, a) LBMs, b) SMs.**

resistance rate to penicillin in *E. coli* isolates from poultry. Schroeder et al. [34] documented lower % resistance rates of 49% for ampicillin and 84% for sulfamethoxazole-trimethoprim. In contrast, Parvin et al. [19] reported significantly higher resistance rates, including 89.5% for ampicillin, 88.4% for sulfamethoxazole-trimethoprim, and 84.9% for tetracycline, which closely aligns with the findings of the present study. The correlation analysis revealed a strong association between phenotypic resistance to ampicillin and the presence of the *bla*TEM gene, indicating that *bla*TEM is a key genetic determinant of ampicillin resistance. This result is consistent with findings in similar studies, where *bla*TEM has been identified as a major contributor to β-lactam antibiotic resistance through the production of TEM-type β-lactamases, which hydrolyse ampicillin effectively [35]. The weak correlation (r = 0.2) between *bla*TEM and *sul*1 or *sul*2 suggests a limited direct association. However, these genes may be present on mobile genetic elements like plasmids or integrons, which promote co-selection under antibiotic pressure. This could account for the observed weak statistical correlation. The low correlations between *bla*TEM and non-AMP antibiotics indicate that *bla*TEM primarily confers AMP resistance. Nevertheless, indirect associations might occur due to co-selection or co-localization with other resistance genes on shared genetic elements. The moderate correlation of *sul*2 with resistance to sulfamethoxazole, a component of SXT, underscores its role in sulfonamide resistance. However, as SXT resistance involves both sulfonamide (e.g., *sul*2) and trimethoprim resistance genes (e.g., *dfr*A), this partial correlation reflects the multifactorial nature of resistance SXT. The AST showed the highest resistance to the combination of sulfamethoxazole-trimethoprim, followed by tetracycline and ampicillin. Li et al. [36] found that 70.9% of isolates were MDR, while only 6.5% exhibited no resistance to the tested antimicrobials. The detection of MDR *E. coli* in this study is concerning and aligns with the findings of Hassan et al. [37], who reported that 100% of poultry samples tested positive for MDR *E. coli*. The rise of MDR pathogens presents a major threat, as these bacteria have the

potential to evolve into 'superbugs,' resulting in treatment failures in poultry and significant public health risks. The widespread occurrence of MDR in poultry is likely due to the overuse of various antibiotic classes, creating intense selection pressure.

Several studies have highlighted poultry farms and their environments, such as litter and wastewater, as major sources of antibiotic residues [30,37]. Moreover, vegetables and animal products from wet markets and shops have been identified as reservoirs of these residues, further contributing to AMR [38,39]. Regular exposure to antibiotic residues accelerates the spread of AMR, particularly in developing countries like Bangladesh. Horizontal transmission of resistant bacteria and genes has also been observed on farms due to antimicrobial residues [40,41]. The improper use of antimicrobials in both human and veterinary medicine plays a significant role in the development of AMR [42]. Lack of knowledge about proper antimicrobial use and the indiscriminate administration of antibiotics are common issues in Bangladesh [43]. Drug sellers and medical representatives often encourage the random use of antimicrobials without prescriptions [44,45]. Inappropriate dosing exposes bacteria to subtherapeutic levels of antimicrobials, enabling them to develop resistance [46]. High doses, conversely, can lead to tissue residues, further promoting resistance [47]. In poultry farming, antimicrobials are frequently administered to entire flocks for infection control and growth promotion [48]. While this reduces mortality and increases profitability, it has serious public health consequences. When humans consume poultry meat containing antibiotic residues, they can inadvertently acquire these residues, contributing to the spread of resistant genes, which are transferred horizontally and vertically among bacterial populations [49]. Although pathogenic bacteria are generally absent in the muscle tissues of healthy birds [50], contamination during slaughter and meat processing can introduce bacteria from the gastrointestinal tract, some of which may already be resistant to antimicrobials due to continuous low-dose exposure.

AMR represents a major threat to both veterinary and public health. In response, the World Health Organization (WHO) has classified antimicrobials into three categories—Access, Watch, and Reserve—based on their significance and the risk of resistance [51]. The Access group is recommended for general use, while the Watch group is reserved for cases where Access group options are ineffective. The Reserve group is intended for use when other options are no longer effective [52]. Addressing AMR requires scientific knowledge and evidence-based practices. Rational use of antimicrobials can improve agriculture while simultaneously reducing the problem of AMR, enhancing both profitability and public health [52,53]. This study is part of a larger project and focused on broiler meat samples collected from LBMs and SMs in the CMA, which may not fully reflect other regions or retail systems in Bangladesh, potentially limiting the generalisability of the findings. The present study does not focus on environmental samples, but the future aim is to investigate the AMR status of SMs and LBMs environmental samples. The AST was conducted using the Kirby-Bauer disc diffusion techniques, which, while widely used, provides qualitative rather than quantitative resistance data, potentially limiting the depth of resistance profiling. The study only investigated resistance to seven antibiotics, leaving the resistance profiles against other clinically relevant antimicrobials unexplored. Molecular characterisation in this study was restricted to detecting the *bla*TEM and *sul*2 genes, while other resistance mechanisms, such as efflux pumps or additional resistance genes, were not examined. The frequency of multidrug-resistant *E. coli* in the human food chain was investigated in this study, which provides important information about potential risks to consumers. This information can guide particular interventions, such as enhanced biosecurity protocols and prudent use of antibiotics in poultry production, processing and distributions.

## Conclusion

This study emphasises the high prevalence of *E. coli* contamination and AMR in broiler meat samples from LBMs and SMs in the CMA, Bangladesh. The findings revealed significant resistance to commonly used antibiotics, including sulfamethoxazole-trimethoprim, tetracycline, and ampicillin, with a substantial proportion of isolates exhibiting MDR. The identification of resistance-associated genes, such as *bla*TEM and *sul*2, further highlights the genetic foundation of AMR in these isolates. These results indicate a pressing need for stricter monitoring of antibiotic use in poultry production and robust food safety measures to mitigate public health risks. Addressing AMR must include integrated strategies involving

policymakers, the poultry industry, and public health stakeholders. This study recommends a large-scale study on AMR patterns of food-producing animals as well as genetic characterization of MDR bacteria covering different geographical areas to mitigate the future risk.

## Supporting Information

**S1 Table. List of sample collected from LBMs and SMs.**
(DOCX)

**S2 Table. Main Dataset.**
(XLSX)

## Acknowledgments

The author sincerely thanks the office staff and lab assistants from the Department of Physiology, Biochemistry, and Pharmacology, as well as the Department of Microbiology and Public Health at Chattogram Veterinary and Animal Sciences University, Chattogram, Bangladesh. The author also appreciates the support of LBM and SM owners for providing the important information and meat samples.

## Author contributions

**Conceptualization:** Hamida Khanom, Mohammad Mahmudul Hassan.

**Data curation:** John I. Alawneh.

**Formal analysis:** Chandan Nath, Mohammad Mahmudul Hassan.

**Funding acquisition:** Mohammad Mahmudul Hassan.

**Investigation:** Md Ridoan Pasha, Mohammad Mahmudul Hassan.

**Methodology:** Hamida Khanom, Chandan Nath, Mohammad Mahmudul Hassan.

**Project administration:** Hamida Khanom, Chandan Nath, Md Ridoan Pasha, Mohammad Mahmudul Hassan.

**Resources:** Philip P. Mshelbwala, Ricardo Soares Magalhaes, John I. Alawneh.

**Software:** Mohammad Mahmudul Hassan.

**Supervision:** Mohammad Mahmudul Hassan.

**Validation:** Philip P. Mshelbwala, Ricardo Soares Magalhaes.

**Writing – original draft:** Hamida Khanom, Mohammad Mahmudul Hassan.

**Writing – review & editing:** Chandan Nath, Philip P. Mshelbwala, Md Ridoan Pasha, Ricardo Soares Magalhaes, John I. Alawneh, Mohammad Mahmudul Hassan.

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
