## [Decision Letter · Decision Letter 0]

4 Mar 2025

PONE-D-25-06137Epidemiology and Molecular Characterisation of Multidrug-Resistant Escherichia coli Isolated from Chicken MeatPLOS ONE

Dear Dr. Hassan,

Thank you for submitting your manuscript to PLOS ONE. After careful consideration, we feel that it has merit but does not fully meet PLOS ONE’s publication criteria as it currently stands. Therefore, we invite you to submit a revised version of the manuscript that addresses the points raised during the review process.

We look forward to receiving your revised manuscript.

Kind regards,

Mabel Kamweli Aworh, DVM, MPH, PhD. FCVSN

Academic Editor

PLOS ONE

Journal Requirements:

Additional Editor Comments : In addition to addressing the reviewers comments please fix the following issues.

Please remove the section numbering from the manuscript.The section should be omitted. Instead, include limitations as the final paragraph in the section.Include line numbers throughout the manuscript to facilitate the review process.In the section, based on your findings, provide a paragraph that includes recommendations or potential future research directions.

Reviewers' comments:

Reviewer's Responses to Questions

**Comments to the Author**

1. Is the manuscript technically sound, and do the data support the conclusions?

Reviewer #1: Yes

Reviewer #2: Yes

Reviewer #3: Yes

2. Has the statistical analysis been performed appropriately and rigorously? 

Reviewer #1: No

Reviewer #2: No

Reviewer #3: Yes

3. Have the authors made all data underlying the findings in their manuscript fully available?

Reviewer #1: Yes

Reviewer #2: No

Reviewer #3: Yes

4. Is the manuscript presented in an intelligible fashion and written in standard English?

Reviewer #1: Yes

Reviewer #2: Yes

Reviewer #3: Yes

5. Review Comments to the Author

Reviewer #1: The authors have conducted a very important and timely study on the epidemiology and molecular characterization of multidrug-resistant Escherichia coli in chicken meat from live bird markets and supermarkets in Chattogram, Bangladesh. This study is highly relevant given the increasing global concerns about AMR and its implications for food safety and public health.

After reviewing this manuscript, i would say it is well-structured with a clearly defined research question. The methodological approach, including antimicrobial susceptibility testing and molecular characterization, was rigorous and appropriate for addressing the study objectives. The authors have also presented their findings in a comprehensive manner, with relevant data supporting their conclusions.

This study provides valuable insights into the burden of MDR E. coli in poultry products in Chattogram, which could help inform surveillance programs and antimicrobial stewardship efforts. The inclusion of both live bird markets and supermarkets strengthens the study by allowing for a comparative analysis, which adds to the novelty of the work.

That said, there are some areas where the manuscript could be improved to enhance clarity, scientific rigor, and overall impact. Below, I provide specific comments and suggestions for improvement.

INTRODUCTION

1. This is a great introduction! However, there is need to include information concerning why you are going to LBM and SMs. Is there any prior evidence differentiating contamination levels or AMR profiles in E. coli from live bird markets vs. supermarkets?

2. Also, your introduction needs to clarify why molecular characterization is necessary (e.g., detection of AMR genes, virulence factors, plasmid-mediated resistance)?

3. The introduction also needs to specify whether similar studies have been conducted in Bangladesh in the past and how this study adds new insights.

4. Since this is the first time this acronym -CMA is used in the main manuscript, please write in full for the first use

METHODS

1. Can you include the particular epidemiological study design that was used in this study (e.g, cross-sectional, observational, experimental)? It is important to include this information so the readers can understand the context of the study

2. Can you indicate a brief justification for selecting this study location? Is it that there have been numerous cases or outbreaks of E.coli around the area or what was the reason for this?

3. It is important to include more information about the study area to give a better context and to allow a more objective interpretation of your findings. I understand that part of this study has been published elsewhere but here you need to provide important details also as not every reader will have the chance to get the already published articles. Please include information concerning

a. The total number of LBM and SM in CMA since this is a prevalence study

b. How the sample size was calculated. How can you justify this sample size of 430 samples?

c. The type (Access, Watch or reserve) and availability of antibiotics in the study area including the policy for antibiotic use among farm animals.

d. A brief description of the sampling technique.

FIGURE 1 - These maps are good as they give the reader more insight to the area. For the map please include

1. The source of the map

2. The cardinal points/North arrow

3. The scale

4. The legend

these are crucial information required for any map in a scientific document

5. Statistical Analysis; if you collected data from LBMs and SM, what tests did you use to make comparisons between LBMs and SMs?

6. Please specify the particular correlation that was done - Spearman, Pearson's, point bi serial e.t.c

RESULTS

1. it would be better to conduct statistical comparisons between LBMs and supermarkets otherwise there maybe little significance in seperating the data from the two.

2. Specify the type of correlation analysis that was performed. As well as the level of significance. were these values statistically significant?

DISCUSSION

1. It is important in the discussion to highlight the public health significance of each major finding as it relates to your objectives. What is the public health significance of a high prevalence of E.coli in broilers?

2. This would be better in the introduction rather than the discussion.

3. it is important you highlight the strengths of your study. Can you provide a brief description of one or two strengths of your study? This is important so we dont only focus on limitations

e. what were the selection criteria for the LBM and SM? These need to be clearly stated

Reviewer #2: It is required that your submitted manuscript should have each line numbered.

It is mandated that page numbers and line numbers be in the manuscript file. Use continuous line numbers (do not restart the numbering on each page).

References: In the text, cite the reference number in square brackets (e.g., “We used the techniques developed by our colleagues [19] to analyze the data”)

Time of study: This study was conducted in 2021 and four years later, it is unclear how the results from this research would still be relevant in the present day.

Introduction

The introduction section does not state what roles the genes associated with or contribute to antibiotic resistance (molecular characterization) play in bacteria carrying them. Also, the authors did not state the rationale for examining breast muscles and the liver in this study.

Methods

2.7 In your methods section you wrote “Correlation analyses between antimicrobial coefficients and corresponding resistance gene abundances”. It is unclear what antimicrobial coefficients mean in this instance.

Did you compare the antimicrobial resistance patterns broadly between the LBMs and SMs to examine if geographical distribution matters?

Results

Kindly note that figures should be at the end of your paper not the result section. PLOS One policy “Do not include figures in the main manuscript file. Each figure must be prepared and submitted as an individual file”

3.2 AMR patterns of E. coli isolates of LBMs and SMs: Is there a reason why the authors did not look for the resistant genes for Tetracycline when a high proportion of 86.78% of isolates showed resistance to Tetracycline (higher than ampicillin)? For ciprofloxacin, the susceptibility in isolates cannot be interpreted as high at 23.14% when compared to the other “highest susceptibility”.

Figure 2c: The correlation coefficient of specific antimicrobials and resistance genes. Did you compute a p-value to determine the significance of the correlation in your samples? The p-value will help in the interpretation of the strength of the relationship between antimicrobial resistance and genes and thus should be included.

References

It is recommended that the authors should update the references to include recent references to reflect current knowledge in this field.

Supplementary Data

The supporting information in this study is sparse thereby making your work hard to reproducible. As explained in PLOS’s Data Policy, be sure to make individual data points and all data and related metadata underlying the findings reported should be provided available as part of the submitted article.

Reviewer #3: The authors are commended for their work on AMR. Once the minor clarifications are made, the paper should be ready for publication

In the methods (there are no line nos to guide), the authors need to state EXACTLY how the liver and breast samples were taken? Was the procedure aseptic. Were the tools reused or single use? Exact size of liver and breast tissue taken? Time interval between slaughter and sample retrieval? Time of the day, etc. It will be good to state what they observed of the environment in which the birds were slaughtered and the chicken were presented prior to sample retrieval across the LBMs.

For the SM samples, we are not told how and from where they were sourced? Same or different suppliers? Same or different packaging? Storage conditions in the SMs? How long have the chicken been placed in the SMs? Etc. Is it not possible that these and other parameters might influence culture yields and trends?

The following statement "The results of this study demonstrated a high prevalence of E. coli in broiler breast muscle and liver samples from both SMs and LBMs, with these isolates" can be better clarified and thus better understood by readers if the above requested but presently missing data is given. Does this statement mean that the birds were prevalence of E. coli is solely from the breast muscle and liver. Or due to post slaughter environmental contamination? Or both? The authors are commended for the limitation section already in place. They may wish to add the lack of environmental samples to the section.

6. PLOS authors have the option to publish the peer review history of their article (what does this mean? ). If published, this will include your full peer review and any attached files.

**Do you want your identity to be public for this peer review?** For information about this choice, including consent withdrawal, please see our Privacy Policy .

Reviewer #1: **Yes: ** Abdulhakeem Abayomi Olorukooba

Reviewer #2: **Yes: ** Damilola Odumade

Reviewer #3: No

---

## [Author Response · Author response to Decision Letter 1]

22 Mar 2025

Dear Editor-in-Chief,

Thank you very much for giving us the opportunity to revise our manuscript (Manuscript ID PONE-D-25-06137) entitled “Epidemiology and Molecular Characterisation of Multidrug-Resistant Escherichia coli Isolated from Chicken Meat” for consideration in “PLOS ONE”. This revision has been made following the valuable comments from the three reviewers. Through you, we would like to sincerely thank the distinguished reviewers for their kind contributions to rendering an improved version of the manuscript from the original submission.

Below are our responses to the reviewers’ and additional editor comments:

Additional Editor Comments: In addition to addressing the reviewers comments please fix the following issues.

• Please remove the section numbering from the manuscript.

Author’s response: Thank you for your comment. We removed the section numbering as per suggestion.

• The Limitations section should be omitted. Instead, include limitations as the final paragraph in the Discussion section.

Author’s response: Thank you for your suggestion. The limitation section have been added with final paragraph in the Discussion section.

• Include line numbers throughout the manuscript to facilitate the review process.

Author’s response: Thank you for your suggestions. We added the line number in the revised manuscript as per suggestion.

• In the Conclusion section, based on your findings, provide a paragraph that includes recommendations or potential future research directions.

Author’s response: Thank you for your suggestion. We have added recommendations for future research.

Responses to reviewers’ comments

Reviewer 1

Comments and Suggestions for Authors

Reviewer #1:

Reviewer comment: The authors have conducted a very important and timely study on the epidemiology and molecular characterization of multidrug-resistant Escherichia coli in chicken meat from live bird markets and supermarkets in Chattogram, Bangladesh. This study is highly relevant given the increasing global concerns about AMR and its implications for food safety and public health.

After reviewing this manuscript, i would say it is well-structured with a clearly defined research question. The methodological approach, including antimicrobial susceptibility testing and molecular characterization, was rigorous and appropriate for addressing the study objectives. The authors have also presented their findings in a comprehensive manner, with relevant data supporting their conclusions.

This study provides valuable insights into the burden of MDR E. coli in poultry products in Chattogram, which could help inform surveillance programs and antimicrobial stewardship efforts. The inclusion of both live bird markets and supermarkets strengthens the study by allowing for a comparative analysis, which adds to the novelty of the work.

That said, there are some areas where the manuscript could be improved to enhance clarity, scientific rigor, and overall impact. Below, I provide specific comments and suggestions for improvement.

Author’s Response: Thank you for your comments and appreciation of our manuscript.

INTRODUCTION

Reviewer comment: 1. This is a great introduction! However, there is need to include information concerning why you are going to LBM and SMs. Is there any prior evidence differentiating contamination levels or AMR profiles in E. coli from live bird markets vs. supermarkets?

Author’s response: Thank you for your comments. Yes, we published articles based on AMR profiles in LBMs and SMs. We have added a published article link for your check. Besides, we have added a line in the introduction addressing LBMs and SMs roles.

https://www.mdpi.com/2079-6382/12/2/418

https://www.mdpi.com/2076-2607/12/12/2535

Reviewer comment: 2. Also, your introduction needs to clarify why molecular characterization is necessary (e.g., detection of AMR genes, virulence factors, plasmid-mediated resistance)?

Author’s response: Thank you for your comments. We have added the necessity of molecular characterization in the introduction part of revised manuscript.

Reviewer comment: 3. The introduction also needs to specify whether similar studies have been conducted in Bangladesh in the past and how this study adds new insights.

Author’s response: We appreciate your feedback and have incorporated these points into the revised manuscript to strengthen the context and significance of our study.

Reviewer comment: 4. Since this is the first time this acronym -CMA is used in the main manuscript, please write in full for the first use

Author’s response: Thank you. Corrected as suggested.

METHODS

Reviewer comment: 1. Can you include the particular epidemiological study design that was used in this study (e.g, cross-sectional, observational, experimental)? It is important to include this information so the readers can understand the context of the study

Author’s response: Thank you. Corrected as suggested.

Reviewer comment: 2. Can you indicate a brief justification for selecting this study location? Is it that there have been numerous cases or outbreaks of E.coli around the area or what was the reason for this?

Author’s response: Thank you so much for your insightful suggestion. We have added a brief justification of study locations.

Reviewer comment: 3. It is important to include more information about the study area to give a better context and to allow a more objective interpretation of your findings. I understand that part of this study has been published elsewhere but here you need to provide important details also as not every reader will have the chance to get the already published articles. Please include information concerning

a. The total number of LBM and SM in CMA since this is a prevalence study

Author’s response: Thank you for your comment. This information have been added as suggested

Reviewer comment: b. How the sample size was calculated. How can you justify this sample size of 430 samples?

Author’s response: Thank you. The sample size calculation formula have been added.

Reviewer comment: c. The type (Access, Watch or reserve) and availability of antibiotics in the study area including the policy for antibiotic use among farm animals.

Author’s response: Thanks for your comments. Corrected as suggested.

Reviewer comment: d. A brief description of the sampling technique.

Author’s response: Thank you. Sampling techniques have been added in the revised manuscript.

Reviewer comment: FIGURE 1 - These maps are good as they give the reader more insight to the area. For the map please include

1. The source of the map

2. The cardinal points/North arrow

3. The scale

4. The legend

these are crucial information required for any map in a scientific document

Author’s response: Thank you for your comment. Figure 1 have been revised accordingly.

Reviewer comment: 5. Statistical Analysis; if you collected data from LBMs and SM, what tests did you use to make comparisons between LBMs and SMs?

Author’s response: Thank you for this comment. We did not perform any significant test, but we did descriptive analysis.

Reviewer comment: 6. Please specify the particular correlation that was done - Spearman, Pearson's, point bi serial e.t.c

Author’s response: Thank you for your comment. We performed Pearson’s correlation.

RESULTS

Reviewer comment: 1. it would be better to conduct statistical comparisons between LBMs and supermarkets otherwise there may be little significance in seperating the data from the two.

Author’s response: Thank you for your nice comment. We conducted Pearson's Chi-squared test for statistical comparisons between LBMs and SMs and the value add in the Table 2 of revised manuscript though it was not significant.

Reviewer comment: 2. Specify the type of correlation analysis that was performed. As well as the level of significance. Were these values statistically significant?

Author’s response: Thank you. We did Pearson’s correlation.

DISCUSSION

Reviewer comment: 1. It is important in the discussion to highlight the public health significance of each major finding as it relates to your objectives. What is the public health significance of a high prevalence of E.coli in broilers? 2. This would be better in the introduction rather than the discussion.

Author’s response: Thank you for your insightful comments. The public health significance have been added in introduction part of revised manuscript.

Reviewer comment: 3. it is important you highlight the strengths of your study. Can you provide a brief description of one or two strengths of your study? This is important so we dont only focus on limitations

Author’s response: Thank you for your insightful comments. We have added a brief description of strengths of our study in the discussion section of revised manuscript.

Reviewer comment: e. what were the selection criteria for the LBM and SM? These need to be clearly stated

Author’s response: Thank you for your comment. LBMs and SMs were conveniently selected

Reviewer #2:

Reviewer comment: It is required that your submitted manuscript should have each line numbered. It is mandated that page numbers and line numbers be in the manuscript file. Use continuous line numbers (do not restart the numbering on each page).

Author’s response: Thank you for your comment. We have added the line number as suggested.

Reviewer comment: References: In the text, cite the reference number in square brackets (e.g., “We used the techniques developed by our colleagues [19] to analyze the data”)

Author’s response: Thank you. Corrected as suggested.

Reviewer comment: Time of study: This study was conducted in 2021 and four years later, it is unclear how the results from this research would still be relevant in the present day.

Author’s response: Thank you for your comment. Following the completion of our project in 2022, we have published several articles based on our findings. However, we believe that this study remains highly relevant, as there is still a lack of published data on this topic within the specific geographical context of Bangladesh. Our research provides valuable insights into antimicrobial resistance trends, which continue to be a critical public health concern.

Introduction

Reviewer comment: The introduction section does not state what roles the genes associated with or contribute to antibiotic resistance (molecular characterization) play in bacteria carrying them. Also, the authors did not state the rationale for examining breast muscles and the liver in this study.

Author’s response: Thank you for your insightful comments. We will revise the introduction to clearly state the roles of the genes associated with antibiotic resistance and how they contribute to resistance mechanisms in E. coli. Additionally, we will include the rationale for selecting breast muscle and liver for examination, emphasizing their relevance in food safety, human consumption, and the potential for bacterial accumulation in different tissues.

Methods

Reviewer comment: 2.7 In your methods section you wrote “Correlation analyses between antimicrobial coefficients and corresponding resistance gene abundances”. It is unclear what antimicrobial coefficients mean in this instance.

Author’s response: Thank you. We deleted the term coefficients for more understanding

Reviewer comment: Did you compare the antimicrobial resistance patterns broadly between the LBMs and SMs to examine if geographical distribution matters?

Author’s response: Thank you. The source of the broiler chickens were almost same for the LBM and SM that’s why we did not consider the geographical distribution.

Results

Reviewer comment: Kindly note that figures should be at the end of your paper not the result section. PLOS One policy “Do not include figures in the main manuscript file. Each figure must be prepared and submitted as an individual file”

Author’s response: Thank you for your comment. Corrected as suggested.

Reviewer comment: 3.2 AMR patterns of E. coli isolates of LBMs and SMs: Is there a reason why the authors did not look for the resistant genes for Tetracycline when a high proportion of 86.78% of isolates showed resistance to Tetracycline (higher than ampicillin)? For ciprofloxacin, the susceptibility in isolates cannot be interpreted as high at 23.14% when compared to the other “highest susceptibility”.

Author’s response: Thank you. We did a separate study where we gave high priority to tetracycline, which was published in the Antibiotic Journal of MDPI.

Reviewer comment: Figure 2c: The correlation coefficient of specific antimicrobials and resistance genes. Did you compute a p-value to determine the significance of the correlation in your samples? The p-value will help in the interpretation of the strength of the relationship between antimicrobial resistance and genes and thus should be included.

Author’s response: Thank you very much. Added as suggested in the revised version.

References

Reviewer comment: It is recommended that the authors should update the references to include recent references to reflect current knowledge in this field.

Author’s response: Thank you. We have updated the reference list.

Supplementary Data

Reviewer comment: The supporting information in this study is sparse thereby making your work hard to reproducible. As explained in PLOS’s Data Policy, be sure to make individual data points and all data and related metadata underlying the findings reported should be provided available as part of the submitted article.

Author’s response: Thank you. Corrected as suggested.

Reviewer #3:

Reviewer comment: The authors are commended for their work on AMR. Once the minor clarifications are made, the paper should be ready for publication

Author’s response: Thank you for your comment.

Reviewer comment: In the methods (there are no line nos to guide), the authors need to state EXACTLY how the liver and breast samples were taken? Was the procedure aseptic? Were the tools reused or single use? Exact size of liver and breast tissue taken? Time interval between slaughter and sample retrieval? Time of the day, etc. It will be good to state what they observed of the environment in which the birds were slaughtered and the chicken were presented prior to sample retrieval across the LBMs.

Author’s response: Thank you. The liver and breast muscle samples were collected using aseptic techniques to prevent contamination. Sterile scalpels and forceps were used to excise the samples, which were immediately placed in sterile containers. Fresh gloves were worn for each sample, and all instruments were sterilized before and after each collection. Samples were collected within 5 minutes of post-slaughter to minimize bacterial changes due to external factors. Sampling was conducted in morning to account for potential variations in handling and storage conditions. Additionally, we will describe the environmental observations at the live bird markets (LBMs), including hygiene practices, slaughter conditions, and how chickens were handled before sample collection. This information will provide a clearer context for potential sources of bacterial contamination.

Reviewer comment: For the SM samples, we are not told how and from where they were sourced? Same or different suppliers? Same or different packaging? Storage conditions in the SMs? How long have the chicken been placed in the SMs? Etc. Is it not possible that these and other parameters might influence culture yields and trends?

Author’s response: Thank you very much. Please see the revised version.

Reviewer comment: The following statement "The results of this study demonstrated a high prevalence of E. coli in broiler breast muscle and liver samples from both SMs and LBMs, with these isolates" can be better clarified and thus better understood by readers if the above requested but presently missing data is given. Does this statement mean that the birds were prevalence of E. coli is solely from the breast muscle and liver? Or due to post slaughter environmental contamination? Or both? The authors are commended for the limitation section already in place. They may wish to add the lack of environmental samples to the section.

Aut

---

## [Decision Letter · Decision Letter 1]

16 Apr 2025

Epidemiology and Molecular Characterisation of Multidrug-Resistant Escherichia coli Isolated from Chicken Meat

PONE-D-25-06137R1

Dear Dr. Hassan,

We’re pleased to inform you that your manuscript has been judged scientifically suitable for publication and will be formally accepted for publication once it meets all outstanding technical requirements.

Kind regards,

Mabel Kamweli Aworh, DVM, MPH, PhD. FCVSN

Academic Editor

PLOS ONE

Additional Editor Comments (optional):

Reviewers' comments:

Reviewer's Responses to Questions

**Comments to the Author**

1. If the authors have adequately addressed your comments raised in a previous round of review and you feel that this manuscript is now acceptable for publication, you may indicate that here to bypass the “Comments to the Author” section, enter your conflict of interest statement in the “Confidential to Editor” section, and submit your "Accept" recommendation.

Reviewer #1: All comments have been addressed

Reviewer #2: All comments have been addressed

Reviewer #3: (No Response)

2. Is the manuscript technically sound, and do the data support the conclusions?

Reviewer #1: Yes

Reviewer #2: Yes

Reviewer #3: Yes

3. Has the statistical analysis been performed appropriately and rigorously? 

Reviewer #1: Yes

Reviewer #2: Yes

Reviewer #3: Yes

4. Have the authors made all data underlying the findings in their manuscript fully available?

Reviewer #1: Yes

Reviewer #2: Yes

Reviewer #3: Yes

5. Is the manuscript presented in an intelligible fashion and written in standard English?

Reviewer #1: Yes

Reviewer #2: Yes

Reviewer #3: (No Response)

6. Review Comments to the Author

Reviewer #1: Authors have addressed all my concerns and comments. I don't have any further questions or comments.

Reviewer #2: Thank you for including the review comment and submitting a great study. Note reference 31 should be corrected (year of publication is missing)

Reviewer #3: The authors seemed to have addressed almost all of the points raised during the first round of the review process.

They added " In the SMs, the sources of chickens were from different farms and LBMs. Usually, SMs collected poultry from different suppliers throughout the years. The broiler carcasses were kept at 40C overnight and sometimes until sold out." However, this statement does not clearly indicate the time interval between slaughter and sample retrieval, time of the day, etc. as was earlier requested. Ditto for slaughter environment observation requested.

"In total, 430 samples were obtained, consisting of 215 liver and 215 muscle samples." from Line 134 should be reported in the results section not under methods.

7. PLOS authors have the option to publish the peer review history of their article (what does this mean? ). If published, this will include your full peer review and any attached files.

**Do you want your identity to be public for this peer review?** For information about this choice, including consent withdrawal, please see our Privacy Policy .

Reviewer #1: **Yes: ** ABDULHAKEEM OLORUKOOBA

Reviewer #2: **Yes: ** Damilola Odumade

Reviewer #3: No

---

## [Editor Report · Acceptance letter]

PONE-D-25-06137R1

PLOS ONE

Dear Dr. Hassan,

I'm pleased to inform you that your manuscript has been deemed suitable for publication in PLOS ONE. Congratulations! Your manuscript is now being handed over to our production team.

Kind regards,

on behalf of

Dr. Mabel Kamweli Aworh

Academic Editor

PLOS ONE